# Simultaneous prediction of antibody backbone and side-chain conformations with deep learning

**Deniz Akpinaroglu**[1], **Jeffrey A. Ruffolo**[2], **Sai Pooja Mahajan**[3], **Jeffrey J. Gray**[2,3]*

**1** Department of Bioengineering, University of California, Merced, CA, United States of America, **2** Program in Molecular Biophysics, The Johns Hopkins University, Baltimore, MD, United States of America, **3** Department of Chemical and Biomolecular Engineering, The Johns Hopkins University, Baltimore, MD, United States of America

* jgray@jhu.edu

**Data Availability Statement:** The source code to train and run DeepSCAb, as well as pretrained models, are available at https://github.com/Graylab/DeepSCAb. The structures predicted by DeepSCAb and alternative methods for

## Abstract

Antibody engineering is becoming increasingly popular in medicine for the development of diagnostics and immunotherapies. Antibody function relies largely on the recognition and binding of antigenic epitopes via the loops in the complementarity determining regions. Hence, accurate high-resolution modeling of these loops is essential for effective antibody engineering and design. Deep learning methods have previously been shown to effectively predict antibody backbone structures described as a set of inter-residue distances and orientations. However, antigen binding is also dependent on the specific conformations of surface side-chains. To address this shortcoming, we created DeepSCAb: a deep learning method that predicts inter-residue geometries as well as side-chain dihedrals of the antibody variable fragment. The network requires only sequence as input, rendering it particularly useful for antibodies without any known backbone conformations. Rotamer predictions use an interpretable self-attention layer, which learns to identify structurally conserved anchor positions across several species. We evaluate the performance of the model for discriminating near-native structures from sets of decoys and find that DeepSCAb outperforms similar methods lacking side-chain context. When compared to alternative rotamer repacking methods, which require an input backbone structure, DeepSCAb predicts side-chain conformations competitively. Our findings suggest that DeepSCAb improves antibody structure prediction with accurate side-chain modeling and is adaptable to applications in docking of antibody-antigen complexes and design of new therapeutic antibody sequences.

## Introduction

Antibodies are specialized proteins that play a crucial role in the detection and destruction of pathogens. The binding and specificity of antibodies are largely determined by the complementarity determining regions (CDRs) that consist of three loops in the light chain and three loops in the heavy chain [1]. Structural diversity is largely achieved by the third loop in the

benchmarking have been deposited at Zenodo: 10.
5281/zenodo.6371490.

**Funding:** This work was supported by National
Science Foundation Research Experience for
Undergraduates grant DBI-1659649 (D.A.),
AstraZeneca (J.A.R.), National Institutes of Health
grants T32-GM008403 (J.A.R.), R35- GM141881
(J.A.R.), R35-GM141881 (J.A.R.), and R01-
GM078221(S.P.M., J.J.G.). Computational
resources were provided by the Maryland
Advanced Research Computing Cluster (MARCC).
The funders had no role in study design, data
collection and analysis, decision to publish, or
preparation of the manuscript.

**Competing interests:** Dr. Gray is an unpaid board
member of the Rosetta Commons. Under
institutional participation agreements between the
University of Washington, acting on behalf of the
Rosetta Commons, Johns Hopkins University may
be entitled to a portion of revenue received on
licensing Rosetta software including methods
discussed/developed in this study. As a member of
the Scientific Advisory Board, J.J.G. has a financial
interest in Cyrus Biotechnology. Cyrus
Biotechnology distributes the Rosetta software,
which may include methods developed in this
study. These arrangements have been reviewed
and approved by the Johns Hopkins University in
accordance with its conflict-of-interest policies.
This does not alter our adherence to PLOS ONE
policies on sharing data and materials."

heavy chain, which determines many antigen binding properties. Additionally, CDR H3 does
not have a canonical fold like the other loops [2], making it challenging to model [3–5]. Cur-
rently, engineering of new antibodies is hindered by accurate prediction of the CDR H3 loop,
including the corresponding side-chains for docking applications. Prediction of side-chains is
a critical component of structure prediction and protein design [6], as the surface of the anti-
body CDR loops including the side-chains play an important role in antigen recognition [7].

There has been a growing interest in effective design of new antibodies since they are com-
monly used in biotherapeutics [8]. Antibody structure determination via techniques like X-ray
crystallography and NMR is challenging and time-consuming. Machine learning methods
improve overall structure prediction and docking [9]. Recently, highly accurate structure pre-
diction models have been proposed for proteins in general [10–12] and for antibodies [13–16].
The performance of AlphaFold2 has been impressively accurate in the recent CASP14 experi-
ment and surpassed most other protein structure prediction methods proposed to date [11].
Unlike the other deep learning-based methods, AlphaFold2 predicts all side-chain rotamers in
addition to the protein backbone. While current antibody prediction methods utilizing deep
learning do not directly predict side-chains, they are all able to predict the backbones with
high accuracy. Hence, a next step towards the advancement of antibody modeling and engi-
neering is the accurate prediction of side-chains to improve overall structure prediction and
docking.

Presently, there are successful methods for rotamer predictions that rely on calculating the
probability of a $\chi$ angle as a function of backbone torsion angles. For instance, SCWRL4 uses
backbone-dependent libraries to calculate rotamer frequencies based on kernel density esti-
mates and kernel regressions [17]. The Rosetta suite employs a similar strategy to repack rota-
mers [18]. Antibody-specific methods like PEARS capture rotameric preferences based on the
immunogenetics numbering scheme to restrict possible side-chain conformations in the sam-
ple space based on positional information [19]. Both SCWRL4 and PEARS require the anti-
body sequence and backbone structure to generate side-chain predictions. They repack the
side-chains onto the provided backbone, and their performance generally declines when the
input is not the crystal backbone. To address these limitations, we propose DeepSCAb (deep
side-chain antibody), a deep neural network that predicts full $F_V$ structures, including side-
chain conformations from only the amino acid sequence.

## Methods

### Antibody structure datasets

**Training dataset.** We used the Structural Antibody Database [20], SAbDab, to curate the
training dataset for DeepSCAb. To ensure only high-quality examples were used for training,
we limited the dataset to structures with 3 Å resolution or better. To assess the impacts of the
sequence redundancy threshold on antibody sequence diversity, we collected structures fil-
tered at a range of sequence identity cutoffs (60%, 70%, 80%, 90%, 95%, 99%), as well as an
unfiltered set of structures. For each set of structures, we calculated the entropy of the amino
acid distribution for each position according to the Chothia numbering (S1 and S2 Figs in S1
File). As expected, we observed a general loss of positional diversity (lower entropy) with
increasing sequence redundancy. However, we observed the opposite trend for the residues
belonging to the CDR H3 loop, with less stringent cutoffs allowing for greater sequence diver-
sity. With this in mind, we selected the 99% sequence identity dataset for model training. For
PDBs with multiple structures (such as crystals with multiple instances of the $F_V$ in the unit
cell), we always select the first. We additionally removed targets belonging to the RosettaAnti-
body benchmark set [21] to evaluate model performance, resulting in a total of 1,433 antibody

structures (64% bound and 36% unbound) for training and validation of our network. Of these structures, a random 95% were used for training and the remaining 5% were used for validation.

## Predicting antibody structure from sequence

DeepSCAb consists of two main components: an inter-residue module for predicting backbone geometries and a rotamer module for predicting side-chain dihedrals. The inter-residue module is initially trained separately and then in parallel with the rotamer module.

**Simultaneous prediction of side-chain and backbone geometries.** The initial layers of the model for predicting pairwise distances and orientations are based on a network architecture similar to that of DeepH3 [13]. The inter-residue module consists of a 3-block 1D ResNet and a 25-block 2D ResNet. As input to the model, we provide the concatenated heavy and light chain $F_V$ sequences, with a total length $L$. The input amino acid sequence is one-hot encoded, resulting in a dimension $L \times 20$. We append an additional binary chain-break delimiter, dimension $L \times 1$, to the input encoding to mark the last residue of the heavy chain. Taken together, the full model input has dimension $L \times 21$. The 1D ResNet begins with a 1D convolution that projects the input features up to $L \times 64$, followed by three 1D ResNet blocks (two 1D convolution with a kernel size of 17) that maintain dimensionality. The output of the 1D ResNet is then transformed to pairwise by redundantly expanding the $L \times 32$ tensor to an $L \times L \times 64$ tensor. Next, this tensor passes through 25 blocks in the 2D ResNet that maintain dimensionality with two 2D convolutions and kernel size of $5 \times 5$. The resulting tensors are converted to pairwise probability distributions over $C_\beta$ distance, $d$, the orientation dihedrals $\omega$ and $\theta$, and the planar angle $\phi$. The inter-residue module is trained as described for DeepH3 [13].

The rotamer module takes as input the inter-residue features. The tensors of dimension $L \times L \times 64$ resulting from the 2D ResNet are transformed to sequential by stacking of rows and columns to obtain a final dimension of $L \times 128$. The rotamer module contains a multi-head attention layer of 1 block with 8 parallel attention heads and a feedforward dimension of 512. The self-attention layer outputs $L \times 128$ tensors, which then pass through a 1D convolution with kernel size of 5. The tensors are converted to rotamer probability distributions that are conditionally predicted for each $\chi$ dihedral using softmax. For example, $\chi_1$ is an input to $\chi_2$, $\chi_1$ and $\chi_2$ are inputs to $\chi_3$, $\chi_1$ through $\chi_3$ are inputs to $\chi_4$, and $\chi_1$ through $\chi_4$ are inputs to $\chi_5$. The predicted rotamers are added back into the inter-residue module: the rotamer tensors are stacked onto the pairwise before the final 2D convolution to update the $d$, $\omega$, $\theta$, and $\phi$ outputs.

Distances are discretized into 36 equal-sized bins in the range of 0 to 18 Å. All dihedral outputs of the network are discretized into 36 equal-sized bins in the range of -180˚ to 180˚ with the exception of $\chi_1$. The $\chi_1$ dihedral is discretized into 36 non-uniform bins, with 6 bins of 30˚ and 30 bins of 6˚. The small bins are centered around -60˚, 60˚, and 180˚, consistent with observed conformational isomers. The planar angle $\phi$ is discretized into 36 equal-sized bins with range 0 to 180˚. Pairwise dihedrals are not calculated for glycine residues due to the absence of a $C_\beta$ atom. side-chain dihedrals were not calculated for glycine and alanine residues due to the absence of a $C_\gamma$ atom and for proline residues due to its non-rotameric nature.

Categorical cross-entropy loss is calculated for each output, where the pairwise losses are summed with equal weight and the rotamer losses are scaled based on each dihedral's frequency of observation: i.e., $\chi_5$ rotamers are much less frequent than $\chi_1$. We do not calculate losses for residue pairs and rotamers missing any of their constitutive atoms, as can occur for poorly resolved flexible regions. The Adam optimizer is used with a learning rate of 0.001. We trained five models with a batch size of 1 on random 95/5 training/validation splits and

averaged over model predictions to generate potentials for downstream applications. DeepS-CAb models were trained on one NVIDIA K80 GPU, which required approximately 100 hours for 120 epochs of training.

**Side-chain only predictions.** To investigate the effect of inter-residual predictions on rotameric predictions, we designed a side-chain-only network as a control. The control network takes as input the one-hot encoded antibody sequence, which passes through a 3 block 1D ResNet. The remaining architecture of the control network as well as its training process is similar to the rotamer module of DeepSCAb (S3 Fig in S1 File). However, there are differences in dimension due to the 1D ResNet returning an $L \times 32$ tensor. The control network models were trained on one NVIDIA K80 GPU, which required 10 hours for 20 epochs of training. We adopted a shorter training process for the control network as the models tended to overfit after 20 epochs.

## Self-attention implementation and interpretation

**Transformer encoder attention layer.** The rotamer module contains a transformer encoder layer that adds the capacity to aggregate information over the entire sequence (S4 Fig in S1 File). We tuned the number of parallel attention heads, the feedforward dimension, and the number of blocks according to validation loss during training. We found that 8 attention heads outperformed 16, feedforward with a dimension of 512 outperformed 1,024 and 2,048, and one block of attention performed identically to two. We further experimented with adding a sinusoidal positional embedding prior to the self-attention layer and obtained identical results, implying that the convolutions in our network contain sufficient information on the order of input elements, rendering positional encoding unnecessary [22].

**Interpreting the attention layer.** In our interpretation of the rotamer attention, we take into consideration only one model out of the five that were trained on random training/validation splits. We do not report an average over the attention matrices from multiple models since they vary amongst themselves (S5 Fig in S1 File). Nevertheless, the properties of attention are conserved across individual models.

We utilize a selected subset of the independent test set to display the most variation across the highly-attended positions as well as the corresponding residue types. This subset consists of the following human PDBs: 1JFQ, 1MFA, 2VXV, 3E8U, 3GIZ, 3HC4, 3LIZ, 3MXW, 3OZ9, and 4NZU.

## Modeling side-chains with DeepSCAb in Rosetta

DeepSCAb generates constraints that are utilized for the prediction of an antibody structure. Discrete potentials are converted to continuous function via the built-in Rosetta spline function. The constraints include all 9 geometries, namely $d$, $\omega$, $\theta$, $\phi$, $\chi_1$, $\chi_2$, $\chi_3$, $\chi_4$, and $\chi_5$. The *ConstraintSetMover* in Rosetta applies these constraints onto the native pose and then the *PackRotamersMover* models side-chain structures. We use the default Dunbrack rotamer library [18] and allow the *PackRotamersMover* to sample extended ranges for $\chi_1$ and $\chi_2$ (using the "-ex1" and "-ex2" flags), as this has been shown to improve side-chain packing performance. We chose the standard *ref2015* full-atom score function with a weight of 1.0 for all constraints. This protocol can repack side-chains on any backbone structure with DeepSCAb predictions.

## Side-chain predictions using alternative methods

To assess the side-chain prediction accuracy against relative solvent accessible surface area, we compare DeepSCAb to three alternative methods: PEARS, SCWRL4, and Rosetta. For each

alternative method, we provide the backbones of the benchmark targets and their sequences. PEARS utilizes antibody-specific rotamer libraries and assigns rotamers based on the IMGT numbering scheme. We generated predictions from PEARS using the publicly available server [19]. SCWRL4 generates $\chi$ kernel density estimates based on backbone-dependent rotamer libraries by minimizing the conformational energies for each residue. We generated SCWRL4 predictions using the SCWRL4.0 algorithm [17]. Rosetta predictions were generated using the same protocol as DeepSCAb, but with only the *ref2015* energy function and no learned constraints.

## Data and code availability

The structures used to train the models presented in this work were collected from SAbDab [20] (http://opig.stats.ox.ac.uk/webapps/newsabdab/sabdab), which curates antibody structures from the Protein Data Bank [23] (https://www.rcsb.org). The source code to train and run DeepSCAb, as well as pretrained models, are available at https://github.com/Graylab/DeepSCAb. The structures predicted by DeepSCAb and alternative methods for benchmarking have been deposited at Zenodo: 10.5281/zenodo.6371490.

## Results

### Overview of the method

Our deep learning method for antibody structure prediction consists of inter-residue and rotamer modules. We trained DeepSCAb to predict antibody backbones as inter-residual distance and orientations. Then, we simultaneously trained the model to predict the side-chain conformations using an attention layer. The pairwise and rotamer probability distributions from DeepSCAb predictions were used for structure realization and packing of the side-chains using Rosetta.

### DeepSCAb predicts inter-residue and side-chain orientations from sequence

DeepSCAb is a neural network that only requires an antibody sequence to predict full $F_V$ structures including side-chain geometries (Fig 1A). The combined sequences of the antibody heavy and light chains are inputted as a one-hot encoding to initially pass through the inter-residue module. The architecture of this module is similar to our previous method for CDR H3 loop structure prediction, DeepH3 (Fig 1B) [13]. The network is pretrained to predict pairwise geometries, such as $d$, $\omega$, $\theta$, and $\phi$. We then feed the penultimate outputs into the rotamer module for the prediction of side-chain conformations, represented as torsion angles $\chi_1$, $\chi_2$, $\chi_3$, $\chi_4$, and $\chi_5$. We explored two strategies for side-chain torsion angle prediction: simultaneous prediction of all angles and conditional prediction of successive torsion angles. For conditional prediction of torsion angles, every $\chi$ angle after $\chi_1$ is predicted given the preceding $\chi$ angle(s) (Fig 1A). After training both model variants, we selected the conditional model as it resulted in lower cross-entropy losses for all nine outputs (inter-residue and rotameric). Within the rotamer module, we included an interpretable self-attention layer before predicting the torsion angles. The predicted side-chain distributions are used to update the inter-residue module to obtain the final inter-residue outputs. We then used Rosetta for full $F_V$ structure realization as well as the repacking of side chains guided by DeepSCAb rotamer constraints (Fig 1C).

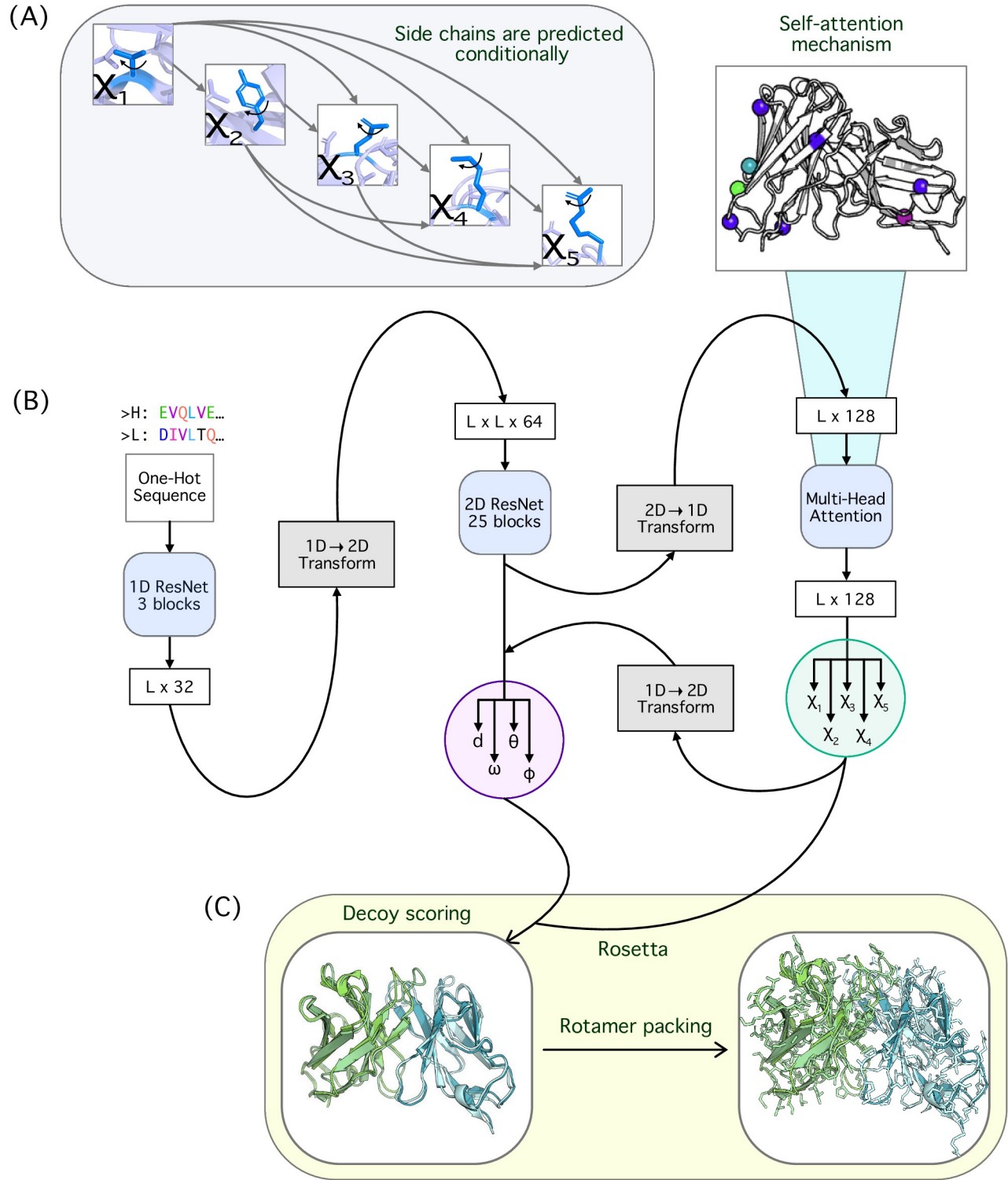

**Fig 1. Overview of the DeepSCAb network architecture.** (A) Conditional side-chain dihedral prediction in DeepSCAb rotamer module with each dihedral after $\chi_1$ depending on previous prediction(s). (B) DeepSCAb architecture for predicting inter-residue geometries and side-chain dihedrals. (C) Applications of DeepSCAb for full $F_V$ realizations and side-chain repacking using Rosetta.

## Rotamer module attends to structurally-conserved anchor positions

The rotamer module includes a self-attention layer that allows us to identify the positions that most significantly influence the side-chain predictions. Rather than attending broadly across the entire antibody sequence, we observed that the model restricted attention to structurally-conserved residues, which we refer to as anchors.

We tested the conservation of anchor positions in various species and settings including ten human antibody targets, a bovine antibody, and mouse and rat sequences with unknown structures. We collected the human antibodies from the independent test set and selected a random bovine antibody (6E9G). Lastly, the mouse and rat antibody structures shown are predictions from DeepSCAb using the protocol described for DeepAb [14], for random paired sequences from OAS [24]. Across the aforementioned range of systems, we found that the anchor positions, as well as anchor residue types, are frequently conserved (Fig 2A).

When we analyzed how attention patterns changed throughout the course of training we observed a process that resembled a search for anchor residues. The initial epochs (before epoch 25) in training are used as a means of scanning for key positions in the sequence. This subsequently results in switching out a few anchors altogether in the beginning stages of training. The high attention residues begin to settle in their positions at epoch 40, however, the ranges of attention assigned remain dynamic up until late epochs. At epoch 100, the model settles on eight anchor positions commonly with highest levels of attention observed (Fig 2B).

## Side-chain predictions improve CDR H3 loop structure accuracy

**Training on backbone geometries improves side-chain predictions.** To assess the side-chain prediction accuracy of the model without any knowledge of backbone preferences, we designed a control network that consists primarily of the DeepSCAb rotamer module. Although the design of our control network would seem to violate the hierarchy of protein

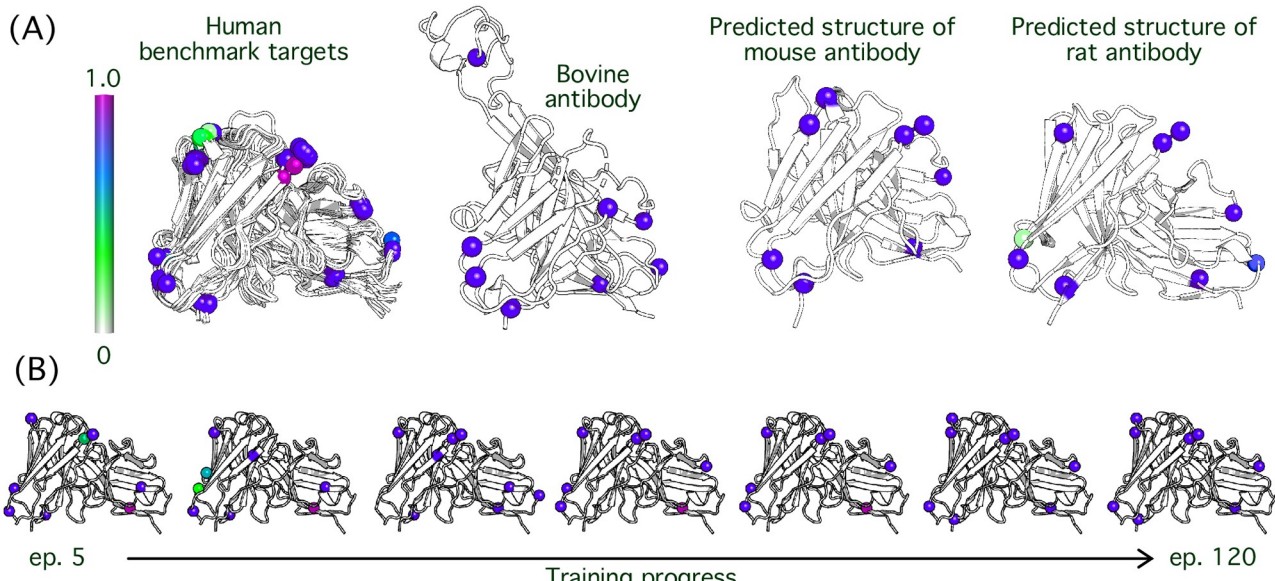

**Fig 2. Identification of anchor residue positions from rotamer module attention.** Rotamer module attention is interpreted to indicate positional significance in side-chain predictions. (A) An attention spectrum (left) ranging from white to magenta represents 0% to 100% attention, respectively. Human, bovine, mouse, and rat antibodies are shown. (B) The variation in attention level is shown with increasing training progress. The epochs represented are 5, 20, 40, 60, 80, 100, and 120.

**Table 1. Decoy discrimination compared to DeepSCAb.**

| Energy | Top 1-Scoring Decoys | | | | Top 5-Scoring Decoys | | | |
|---|---|---|---|---|---|---|---|---|
| | **Better** | **Same** | **Worse** | **<RMSD>** | **Better** | **Same** | **Worse** | **<RMSD>** |
| DeepSCAb | - | - | - | 3.2Å ± 1.3 | - | - | - | 2.5Å ± 1.2 |
| Control | 32 | 7 | 10 | 5.0Å ± 3.8 | 23 | 11 | 15 | 3.3Å ± 2.4 |
| DeepH3 | 10 | 33 | 7 | 3.2Å ± 1.4 | 10 | 35 | 4 | 2.6Å ± 1.3 |

structure, in which the local tertiary environment is a critical determinant of side-chain conformation, we sought to investigate the capacity of a ResNet model to infer side-chain conformations from sequence position alone. This control is similar in principle to the PEARS method [19], which uses positional statistics collected for IMGT-numbered positions to predict side-chain conformations.

We evaluated the control network and the full DeepSCAb on a decoy discrimination task using a set of structures generated by Jeliazkov et al. [25], for the RosettaAntibody benchmark with 2,800 decoys per target. In the decoy discrimination task, we evaluate the ability of an energy function, such as the rotameric distributions predicted by DeepSCAb, to distinguish near-native conformations from a large set of alternative conformations (decoys). For each target in the benchmark, we score each of the decoys using the control network and DeepSCAb, and compare the decoy ranking capacity of the models by measuring the RMSD from the native for the top-1 and top-5 scoring structures. For the top-1 scoring structures, DeepSCAb (RMSD = 3.2 Å) outperformed the control network (RMSD = 5.0 Å) by 1.8 Å (32 better, 7 same, 10 worse). Among the top-5 scoring structures, DeepSCAb (RMSD = 2.5 Å) outperformed the control network (RMSD = 3.3 Å) by 0.8 Å (23 better, 11 same, 15 worse) (Table 1). Due to the considerable improvement observed in DeepSCAb over the control network, we conclude that direct injection of structural priors, through prediction of the inter-residue geometries, is beneficial for antibody side-chain predictions.

**Training on side-chain geometries improves inter-residue predictions in return.** Since DeepSCAb outperformed the side-chain-only control network, we next evaluated the impacts of learning side-chain conformations on pairwise residue-residue geometry predictions. First, we compared the cross-entropy loss achieved by DeepSCAb to that of DeepH3 for the trained ensembles (S6 Fig in S1 File). For every pairwise geometry prediction, DeepSCAb achieved lower loss than DeepH3 for both the training and validation datasets, suggesting that side-chain prediction can improve prediction of inter-residue geometries. Given this improvement, we next compared the performance of DeepSCAb to DeepH3 on the decoy discrimination task. For the Top 1-scoring decoys, DeepSCAb modestly outperformed DeepH3 (10 better, 33 same, 7 worse; <ΔRMSD> = 0 Å). For the Top 5-scoring decoys, DeepSCAb outperformed DeepH3 (10 better, 35 same, 4 worse; <ΔRMSD> = −0.1 Å) (Table 1).

Using the independent test set, we plotted the structures chosen by DeepH3 against the ones chosen by DeepSCAb based on their RMSD (Å) (Fig 3). DeepSCAb was better at distinguishing near-native structures in both Top 1 and Top 5 plots, though improvements over DeepH3 were most notable in the Top 5 comparison (Fig 3A). We then analyzed two targets chosen from the Top 5 decoys for the three methods and *ref2015* (Rosetta energy). We show the 2,500 structure scores against RMSD in the CDR H3 loop for the target 2FB4 (loop length of 19) (Fig 3B). DeepSCAb outperformed the control network (ΔRMSD = −10.8 Å), *ref2015* (ΔRMSD = −10.2 Å), and DeepH3 (ΔRMSD = −1.2 Å). Comparison of the structures identified by DeepSCAb and DeepH3 revealed that both models are able to place the CDR H3 loop in the correct orientation, however, the addition of side-chain information in DeepSCAb

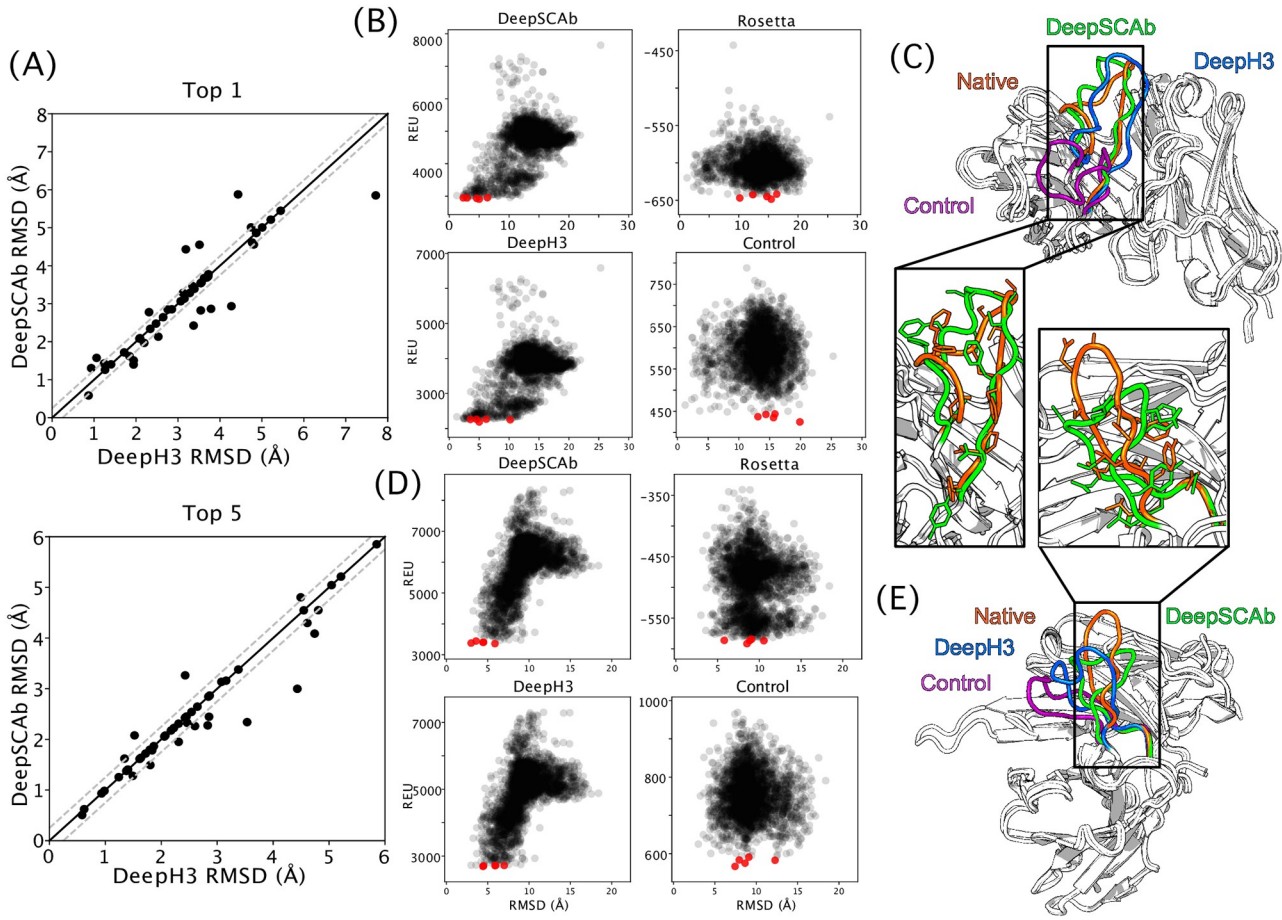

**Fig 3. Comparison of CDR H3 structure prediction accuracy.** The accuracy with which DeepSCAb, DeepH3, and the control network predict the CDR H3 loop structure is measured via decoy structure scoring tasks. (A) In Top 1-scoring decoy structures (top) and Top 5-scoring structures (bottom), the performance of DeepSCAb is compared to DeepH3 using the test set. (B) The CDR H3 energies for the three methods and Rosetta (ref2015) that correspond to 2FB4 are plotted against their RMSD. The five best scoring structures for each plot are indicated in red. (C) The best prediction from Top 5-scoring decoys for target 2FB4 are shown for DeepSCAb (green, 2.34 Å RMSD), DeepH3 (blue, 3.535 Å RMSD), and the control network (purple, 13.091 Å RMSD) all compared to the native (orange). (D) The CDR H3 energies for the three methods and Rosetta (ref2015) that correspond to 3MLR are plotted against their RMSD. The five best scoring structures for each plot are indicated in red. (E) The best prediction from Top 5-scoring decoys for target 3MLR are shown for DeepSCAb (green, 2.998 Å RMSD), DeepH3 (blue, 4.432 Å RMSD), and the control network (purple, 7.441 Å RMSD) all compared to the native (orange).

results in a more accurate structure (Fig 3C). We further plotted the funnel energies in the CDR H3 loop for the target 3MLR (loop length of 17) (Fig 3D). DeepSCAb outperformed the control network ($\Delta$RMSD = −4.4 Å), *ref2015* ($\Delta$RMSD = −5.8 Å), and DeepH3 ($\Delta$RMSD = −1.4 Å). We superimposed predicted structures from each method and the native for the target 3MLR, one of the longer and more difficult of the CDR H3 loops (Fig 3E). DeepSCAb predicts the loop structure with the highest accuracy. Hence, the addition of side-chain orientations is beneficial for accurately predicting pairwise geometries.

## DeepSCAb is competitive with alternative rotamer packing methods

The context of the predicted side chains is crucial in determining the accuracy and usefulness of a method. Side chains that are exposed to a solvent play an active role in the binding of an

antigen, yet are also inherently the most flexible. We evaluated the performance of our method and three alternative methods as a function of relative side-chain solvent accessible surface area (SC SASA) using the Rosetta *rel_per_res_sc_sasa* method, normalizing using reference SASAs from Tien et al. [26]. We compared the success of DeepSCAb in predicting side-chain conformations to PEARS, SCWRL4, and Rosetta, using the native structure as a reference for all measurements. We omitted the target 3MLR from side-chain packing and relative SC SASA comparisons as PEARS was unable to model this structure due to its long L3 loop.

The exposure of side-chains to solvent (SC SASA) is a key determinant of whether a computational method can be expected to accurately recover the native side-chain conformation. In Fig 4A, we compared the repacking performance of DeepSCAb to alternatives and found that DeepSCAb produced competitive side-chain packing results for buried residues, or a relative SC SASA of 0, and across a range of increasing solvent exposures (S1 Table in S1 File). With increasing solvent exposure, we see a consistent degradation of performance for all methods. This is expected, as the side chains gain additional conformational freedom with increasing solvent exposure, making accurate predictions increasingly challenging.

A key distinction between DeepSCAb and alternative methods is that its learned side-chain potentials depend only on the antibody sequence. As a result, the predicted rotameric distributions are based on an implicit backbone learned by the inter-residue module. When this implicit model is incorrect, we expected the side-chain predictions to be less accurate as well. To test this hypothesis, we generated backbones from the DeepSCAb pairwise predictions using the structure realization procedure proposed for DeepAb [14]. Then, we quantify the error in this DeepSCAb backbone as the deviation from native of the $C_\beta$ atoms when the framework residues are aligned. After packing side-chains for these predicted backbones, we measure the cosine distance from the native dihedral for $\chi_1$-$\chi_4$ ($\chi_5$ is omitted due to limited data). We compare the backbone error to side-chain dihedral error and find that as the DeepSCAb-predicted backbone becomes less accurate (higher $C_\beta$ deviation), the side-chain dihedral errors increase (Fig 4B).

## Discussion

The results show that our method is a step towards accurate antibody structure prediction via inclusion of side-chain conformations. We demonstrated that DeepSCAb predictions remain competitively accurate at varying side-chain surface exposure. In investigating the causes of failed side-chain predictions, we found that DeepSCAb rotamer module performance is dependent on the quality of its inter-residue geometry predictions. Thus, as methods for protein backbone prediction (and simultaneous side-chain prediction, as with AlphaFold2 [11]) continue to improve, it will be less important to predict side-chains separately. In the meantime, our method complements existing methods for antibody structure prediction.

Using the rotamer module attention, we are able to identify the residue types and positions that are the most influential in the context of side-chain predictions. While this analysis provides insight into rotamer prediction, it does not reveal local biophysical interactions that could be tied into the fundamentals of side-chain conformation in complex energy landscapes. The anchor sites are consistently scattered throughout the sequence, in stark contrast with the local chemical environment typically considered by most side-chain placement algorithms. Perhaps DeepSCAb is learning an internal, structurally conserved numbering scheme as a reference for side-chain prediction similar to the well-performing PEARS algorithm [19], which was designed to use antibody-specific residue positioning to condition rotamer predictions. Alternatively, the model could be identifying global antibody features such as germline class or species, which have some side-chain conformations conserved.

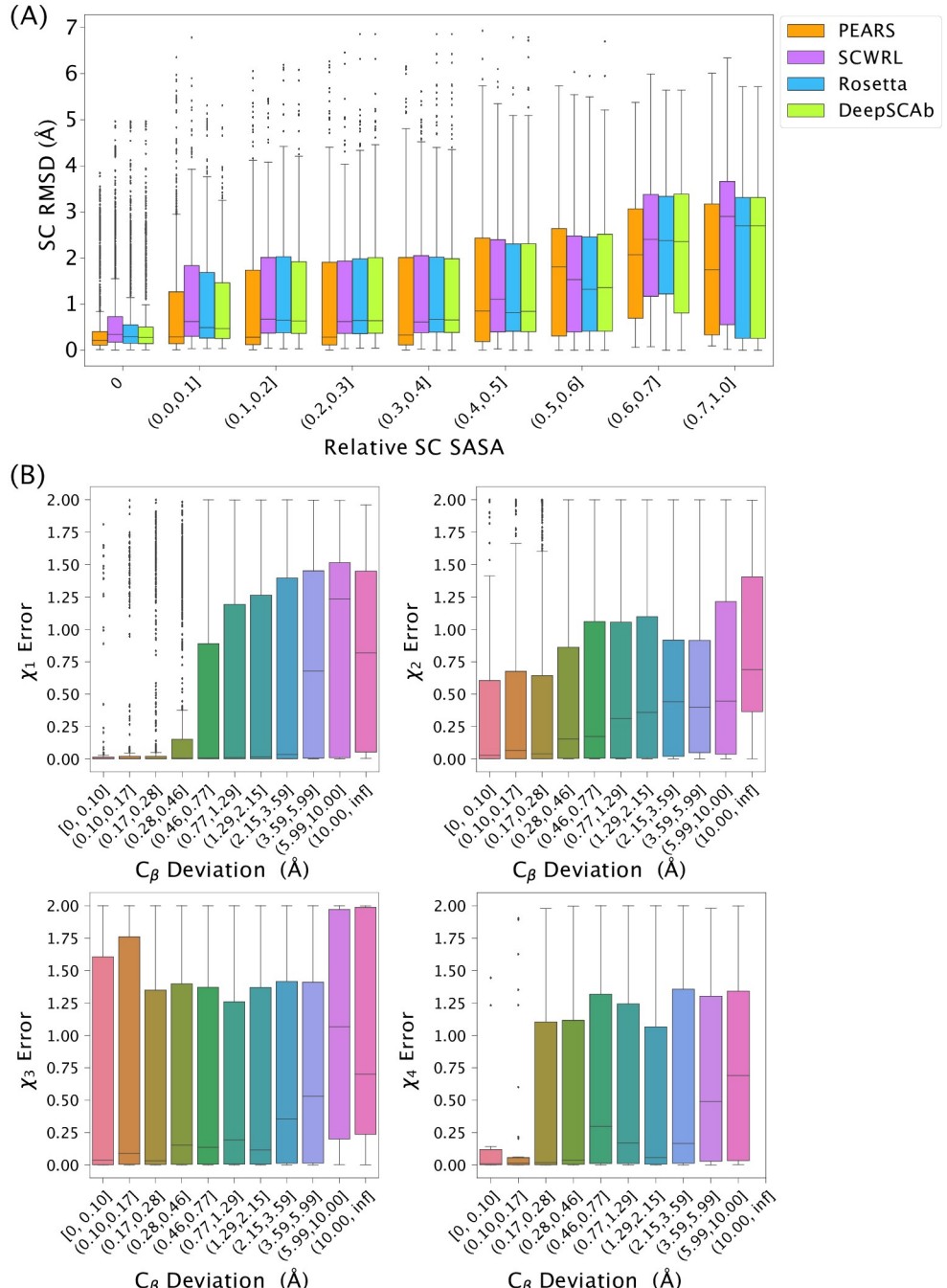

**Fig 4. Impacts of solvent exposure and learned backbone error on side-chain prediction accuracy.** (A) Comparison of repacked side-chain RMSD for PEARS, SCWRL, Rosetta, and DeepSCAb with increasing relative side-chain solvent accessible surface area (SC SASA). (B) Comparison of error in DeepSCAb-predicted backbones versus error in side-chain dihedral prediction. Backbone error is measured as the $C_\beta$ deviation between the predicted structure and the native when the framework residues are aligned. Side-chain dihedral error is measured as a cosine distance between the predicted and native dihedrals.

It is well-established that access to backbone context improves side-chain predictions [18], which is supported by our results. Additionally, we show that inclusion of side-chains enables structure prediction models to more effectively predict pairwise geometries (*i.e.*, lower loss). We found that informing the model of rotameric outputs improved the ability of our model to discriminate near-native CDR H3 loop structures. As this improvement is limited, rather than a significant overall improvement in predictions, we believe that the model is reducing its loss by more confidently predicting pairwise geometries that were already correct. Thus going forward we must consider an implementation of side chain learning that is tailored to pairwise geometries that the model is unable to predict correctly by itself. Concurrent with this work, improved methods for antibody structure prediction have been developed: DeepAb [14] uses a similar architecture to predict inter-residue geometries, and ABlooper [15] predicts CDR loop coordinates directly. Our work suggests that both of these methods might be improved by incorporating side-chain context into predictions.

Most side-chain repacking methods sample conformations based on a backbone-dependent rotamer library [17], and the accurate PEARS method for antibody side-chain repacking estimates $\chi$ angle densities based on a position-dependent rotamer library [19]. Since our method does not require structure as an input, DeepSCAb should be more robust to changes in backbone structure for cases where the model's implicit backbone (e.g., the inter-residue predictions) is close to correct. This feature is useful when there are multiple potential backbone conformations of interest [27], e.g., for the design of new therapeutic antibodies. Another deep learning method that conditionally samples rotamers has been proposed for protein sequence design by Anand et al. [28], which predicts rotamers given the native residue type for the fixed backbone. While rotamer prediction accuracy is higher with the availability of the native backbone structure, the ability to predict in its absence renders DeepSCAb uniquely useful. With minimal modification, our network can aid antibody design. For instance, DeepSCAb can be used in parallel with RosettaAntibodyDesign [29] for rapid placement of side-chains or to hallucinate new antibody sequences using the trRosetta architecture [10].

## Conclusion

In this study, we investigated the effect of inter-residual predictions on the accuracy of side-chain dihedrals as well as the effect of rotamer predictions on the overall antibody structure prediction accuracy. We found that DeepSCAb competitively predicts rotamers when compared to alternative methods that require true backbone coordinates. The performance of our method is robust to when the backbone is perturbed or deviates from the crystal structure. Since DeepSCAb predicts a probability distribution over the backbone and side-chain geometries, we expect it will be adaptable to and useful for designing new antibodies.

## Supporting information

**S1 File. Supporting information for manuscript.** File containing containing supporting tables and figures for DeepSCAb method for prediction of antibody side-chain conformations. (PDF)

## Acknowledgments

We thank the Gray Lab for helpful discussions and advice. Dr. Gray is an unpaid board member of the Rosetta Commons. Under institutional participation agreements between the University of Washington, acting on behalf of the Rosetta Commons, Johns Hopkins University may be entitled to a portion of revenue received on licensing Rosetta software including

methods discussed/developed in this study. As a member of the Scientific Advisory Board, J.J. G. has a financial interest in Cyrus Biotechnology. Cyrus Biotechnology distributes the Rosetta software, which may include methods developed in this study. These arrangements have been reviewed and approved by the Johns Hopkins University in accordance with its conflict-of-interest policies.

## Author Contributions

**Conceptualization:** Deniz Akpinaroglu, Jeffrey A. Ruffolo, Sai Pooja Mahajan, Jeffrey J. Gray.

**Funding acquisition:** Jeffrey J. Gray.

**Methodology:** Deniz Akpinaroglu, Jeffrey A. Ruffolo, Sai Pooja Mahajan, Jeffrey J. Gray.

**Resources:** Jeffrey J. Gray.

**Software:** Deniz Akpinaroglu, Jeffrey A. Ruffolo.

**Supervision:** Jeffrey J. Gray.

**Writing – original draft:** Deniz Akpinaroglu.

**Writing – review & editing:** Jeffrey A. Ruffolo, Sai Pooja Mahajan, Jeffrey J. Gray.

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
