## [Decision Letter · Decision Letter 0]

13 Dec 2021

PONE-D-21-28666Improved antibody structure prediction by deep learning of side chain conformationsPLOS ONE

Dear Dr. Gray,

Thank you for submitting your manuscript to PLOS ONE. After careful consideration, we feel that it has merit but does not fully meet PLOS ONE’s publication criteria as it currently stands. Therefore, we invite you to submit a revised version of the manuscript that addresses the points raised during the review process.

All reviewers appreciated the presented work and its importance to the field. However, both major and minor issues were identified that need to be addressed in the revised manuscript. Please refer to the reviewers` comments at the end of this letter for details.

As an academic editor, I also have a number of comments and requests to update the manuscript in order to improve its clarity and reproducibility as well as to make it compliant with the journal publication policy.

Scientific issues

1. Methods section does not contain any information about validation/control sets.

2. The input vector is not fully defined. What is L: the length of the entire antibody, variable region of both chains, or CDR part only? L as an input assumes a constant value, but all antibodies, especially in Fv part only, are of variable length. When a shorter sequence is considered, how the empty positions are filled, padded with 0s or something else? If the input is in one-hot format, why the chain delimiter occupies only one position, not 2? What is the input format for the 21st position (light vs heavy chain) used then?

3. 99% sequence identity threshold appears very permissive that may result in the over-optimistic results. The authors need to compute and provide distributions of the antibodies per bins of sequence identity, using e.g. USEARCH/UCLUST tool, followed by the distribution as to where most of those mismatches fall into for each sequence identity bin (e.g., in the CDR or otherwise).

4. While the authors claim importance of the predictions to the subsequent AB-antigen docking, they provided no information as to how antigens and induced fit were accounted for in their model. Furthermore, the authors need to discuss and describe the following aspects of training and validating of predicted structural data: (1) Report the number of AB structures in bound and unbound forms used in training and test sets. (2) Given that many loop regions (which are important in the context of CDR) are frequently too flexible to be resolved by X-ray or result in multiple occupancies in the ATOM section of PDB file, the authors have to describe how they used structures with missing atoms or atoms with multiple occupancies. The same pertains to NMR-based modes – which model from the ensemble was used for training and validation?

5. Methods should contain information how values of relative solvent accessibility were computed. Figure 4 contains ranges with SASA > 100%, which is confusing.

6. If not demonstrated in Results, the issue of cross-reactivity of Abs should be at least pointed out in Discussion. There are many instances, e.g. in autoimmune disorders, when the same Ab naturally evolved against viral proteins cross-reacts with the human (host in general) proteins that have no sequence similarity to the viral antigens. For example, use published reports in lupus. The authors at least need to offer some hypothesis in Discussion as to why this may happen.

Editorial issues

1. Description of the Training set appears to be copied verbatim from the authors` previous publication, which technically falls into self-plagiarism category.

2. All references have to adhere to the scientific citation format. For example, references 14, 15, and 16 do not contain the source of publication, such as journal.

3. All associated software should be made publicly available in order to allow reviewers to assess its functionality.

We look forward to receiving your revised manuscript.

Kind regards,

Alexey Porollo, PhD

Academic Editor

PLOS ONE

Journal Requirements:

2. Please amend your Methods section to provide a URL for the Protein Data Bank.

“This work was supported by National Science Foundation (https://nsf.gov/) Research Experience for Undergraduates grant DBI-1659649 (D.A.), AstraZeneca (https://www.astrazeneca.com/) (J.A.R.), National Institutes of Health (https://www.nih.gov/) grants T32-GM008403 (J.A.R.), R01-GM078221(S.P.M., J.J.G.), and R01-GM127578(S.P.M., J.J.G.). Computational resources were provided by the Maryland Advanced Research Computing Cluster (MARCC) (https://www.marcc.jhu.edu/).”

“Dr. Gray is an unpaid board member of the Rosetta Commons. Under institutional participation agreements between the University of Washington, acting on behalf of the Rosetta Commons, Johns Hopkins University may be entitled to a portion of revenue received on licensing Rosetta software including methods discussed/developed in this study. As a member of the Scientific Advisory Board, J.J.G. has a financial interest in Cyrus Biotechnology. Cyrus Biotechnology distributes the Rosetta software, which may include methods developed in this study. These arrangements have been reviewed and approved by the Johns Hopkins University in accordance with its conflict-of-interest policies.”

“This work was supported by National Science Foundation Research Experience for Undergraduates grant DBI-1659649 (D.A.), AstraZeneca (J.A.R.), National Institutes of Health grants T32-GM008403 (J.A.R.) and R01-GM078221(S.P.M., J.J.G.). Computational resources were provided by the Maryland Advanced Research Computing Cluster (MARCC).”

“This work was supported by National Science Foundation (https://nsf.gov/) Research Experience for Undergraduates grant DBI-1659649 (D.A.), AstraZeneca (https://www.astrazeneca.com/) (J.A.R.), National Institutes of Health (https://www.nih.gov/) grants T32-GM008403 (J.A.R.), R01-GM078221(S.P.M., J.J.G.), and R01-GM127578(S.P.M., J.J.G.). Computational resources were provided by the Maryland Advanced Research Computing Cluster (MARCC) (https://www.marcc.jhu.edu/).”

Reviewers' comments:

Reviewer's Responses to Questions

**Comments to the Author**

1. Is the manuscript technically sound, and do the data support the conclusions?

Reviewer #1: Yes

Reviewer #2: Yes

Reviewer #3: Yes

2. Has the statistical analysis been performed appropriately and rigorously? 

Reviewer #1: Yes

Reviewer #2: No

Reviewer #3: Yes

3. Have the authors made all data underlying the findings in their manuscript fully available?

Reviewer #1: No

Reviewer #2: No

Reviewer #3: Yes

4. Is the manuscript presented in an intelligible fashion and written in standard English?

Reviewer #1: Yes

Reviewer #2: Yes

Reviewer #3: Yes

5. Review Comments to the Author

Reviewer #1: This is a very decent piece of work!

The authors develop a ML pipeline that predicts not only antibody structure from its sequence, but also the side chain conformations. Of note, predicting antibody structure is still a very challenging task for which Alphafold2 has shown disappointing results (probably due to the lack of co-evolutionary information between antibodies and antigens). Therefore, there is critical need for such type of tools.

Particularly interesting points:

- The authors relax predicted atomic/residue distances into a realistic 3D structure using Rosetta, which is therefore more useful and makes comparison to experimental structures easier.

- This study extends their previous work (Ref 13) by additionally prediting the side chain of the antibody, which is indeed a lacking point of most predictions methods. As they write, Alphafold2 does predict side residue conformation (but with low accuracy), so a tool specifically benchmarked for antibodies is needed. Abodybuilder does predict it though.

- The authors use the attention layer as to interprete the results (which anchors are most determinant in the side chains conformations), which is a good example of gained knowledge/interpretability from a trained model, which is appreciated.

- We believe the authors provided convincing sets of controls as to show the preformance of this tool.

Major point:

- We recall that AbodyBuilder also predicts side-chains in two ways: "complete" prediction, where every side chain is predicted (using PEARS, we think), and "partial" prediction, where side chains of identical residues from the template are retained, and the remaining side chains are also predicted. Wouldn’t it make sense to compare the performance to AbodyBuilder (and not only PEARS alone), or did we miss something?

Weak points, that are minor but would improve the manuscript.

- In case there is a revision round, please make the code and data available to the reviewers. We couldn’t assess whether it is easy to use/reproducible.

- The language is pretty technical, and some concepts could be better explained to non-specialists, such as the interest of usnig «decoy discrimination», the conditional prediction of side chains in Figure 1A.

- The discussion starts with the importance of the work in the context of docking, but is not really substantiated, although it is likely true... Could the authors discuss more reasons to believe so? For instance, evil’s advocate could say that due to side chain flexibility, knowing the side chains might or might not help docking that much.

- We didn’t understand whether the rotamer libraries were inputed, and from which data. How do the rotamers in the final predicted structures differ from the used rotamer library? Does it advocate for antibody-specific rotamer libraries?

Reviewer #2: In their paper 'Improved antibody structure prediction by deep learning of side chain conformations' authors present DeepSCAb - novel deep learning

method of predicting structure of antibody's variable fragments from sequence. The authors use complicated, highly tailored for the task model that combines 1D and 2D residual

convolutional blocks with multi-head attention module to predict dihedral angles for amino acid side chains.

In my opinion, the paper clearly explains the development of the method, performs several analyses of its output and compares its performance to other methods.

This work is of great importance for the development of new powerful biotherapeutics. I especially like how the authors predict side chain dihedral values conditionally.

However, I have the following concerns regarding this publication:

Major Comments:

1. My major concerns relate to evaluation of the model's performance and comparisons to what the authors call 'control network' and DeepH3 (previous work by the authors).

First, I don't see how the idea behind the control network makes sense. The authors train it to predict side chain conformations from sequence without any information about

backbone, which in my opinion is meaningless - it just violates the hierarchy of protein structure and clearly calls for much more advanced modeling techniques such as Alphafold,

which first can internally infer backbone conformation and then predict side chains, making this task essentially equal to the full protein structure prediction. Therefore,

I don't see how this network is useful as a baseline.

Second, the performance improvement compared to DeepH3 is very modest, if present at all (the authors give dRMSDs of 0 and -0.1 angstroms). I would like to see uncertainties of all the

RMSD and dRMSD values presented in the Table 1 and in corresponding parts of the text. This will allow to see if the performance improvements are statistically significant.

In the introduction, the authors cite a number of works which present other methods for antibody structure prediction (e.g. ABLooper) and general protein structure prediction

(e.g. Alphafold2 and RoseTTAFold). I think that the authors should use those methods and compare their performance to DeepSCAb's.

2. The authors used PEARS, SCWRL4 and Rosetta for side chain prediction. I think that there is not enough detail given about settings used for Rosetta, which allows, in addition

to the force field, tune the number of rotamers used as initial seed and other parameters, which significantly change performance. For example, there are settings that allow Rosetta

to use more rotamers during packing. Did the authors explore that? Also, there are newer methods of side chain packing that claim to be better than the ones used by the authors.

Minor comments:

1. Would be also interesting to discuss cases when DeepSCAb is worse then other methods. Why do you think this happens?

2. In the method description would be helpful to state explicitly what L is - is it the length of the full protein or just the length of the loop? I presume you model

full protein, but do you think it could be interesting (and computationally less expensive) to model only part of anybody having most of it fixed?

3. Error bars in Figure 4 should not reach negative RMSD values, they should be cut off at 0. I also not sure about usefulness of that figure given that the difference between all the

method is insignificant.

Reviewer #3: The manuscript presents a method that explicitly addresses the antibody side chain modeling. Previous approaches focused on backbone modeling only, this is the first antibody modeling approach that predicts side chains. This is done elegantly by adding a new module to the antibody network that predicts backbone distances and angles for further optimization by Rosetta. The new module predicts the side chain rotamer angles depending on the prediction of backbone distances and angles.

Comments:

1. Test set cutoff of 99% can lead to very similar H3 loops in the training and test set.

2. Generation of 2,800 decoy models - how much time this takes for one antibody sequence? Can the same results be achieved with fewer decoys?

3. Stated that "the addition of side chain orientations is beneficial for accurately predicting pairwise geometries." but there is no significant improvement from DeepH3 in Backbone RMSD (top1 and top 5).

6. PLOS authors have the option to publish the peer review history of their article (what does this mean?). If published, this will include your full peer review and any attached files.

Reviewer #1: No

Reviewer #2: No

Reviewer #3: No

---

## [Author Response · Author response to Decision Letter 0]

18 Apr 2022

Please see full response to reviewers in attached file.

---

## [Decision Letter · Decision Letter 1]

25 May 2022

Simultaneous prediction of antibody backbone and side-chain conformations with deep learning

PONE-D-21-28666R1

Dear Dr. Gray,

We’re pleased to inform you that your manuscript has been judged scientifically suitable for publication and will be formally accepted for publication once it meets all outstanding technical requirements.

Kind regards,

Alexey Porollo, PhD

Academic Editor

PLOS ONE

Additional Editor Comments (optional):

Reviewers' comments:

Reviewer's Responses to Questions

**Comments to the Author**

1. If the authors have adequately addressed your comments raised in a previous round of review and you feel that this manuscript is now acceptable for publication, you may indicate that here to bypass the “Comments to the Author” section, enter your conflict of interest statement in the “Confidential to Editor” section, and submit your "Accept" recommendation.

Reviewer #1: All comments have been addressed

Reviewer #2: All comments have been addressed

Reviewer #3: (No Response)

2. Is the manuscript technically sound, and do the data support the conclusions?

Reviewer #1: Yes

Reviewer #2: Yes

Reviewer #3: Yes

3. Has the statistical analysis been performed appropriately and rigorously? 

Reviewer #1: Yes

Reviewer #2: Yes

Reviewer #3: Yes

4. Have the authors made all data underlying the findings in their manuscript fully available?

Reviewer #1: Yes

Reviewer #2: Yes

Reviewer #3: Yes

5. Is the manuscript presented in an intelligible fashion and written in standard English?

Reviewer #1: Yes

Reviewer #2: Yes

Reviewer #3: Yes

6. Review Comments to the Author

Reviewer #1: The authors have made the text clearer and answered all my points, including the points of the other reviewers.

Reviewer #2: I believe that the authors have addressed all major concerns raised in the previous round of the review.

Reviewer #3: The authors revised and improved the manuscript based on the editor and reviewers comments

Minor comments

Using 99% sequence identity cut-off: I don’t think that the entropy analysis is the best approach in this case. The more straightforward solution is to show that comparable results can be achieved for other cut-offs, for example 90% and 95%.

Is it possible to add the runtimes for antibody modeling, including training of the network, inference, and RosettaAntibody modeling.

7. PLOS authors have the option to publish the peer review history of their article (what does this mean?). If published, this will include your full peer review and any attached files.

Reviewer #1: No

Reviewer #2: No

Reviewer #3: No

---

## [Editor Report · Acceptance letter]

6 Jun 2022

PONE-D-21-28666R1 

Simultaneous prediction of antibody backbone and side-chain conformations with deep learning 

Dear Dr. Gray:

I'm pleased to inform you that your manuscript has been deemed suitable for publication in PLOS ONE. Congratulations! Your manuscript is now with our production department. 

Kind regards, 

on behalf of

Dr. Alexey Porollo 

Academic Editor

PLOS ONE